# Loss of *Cnot6l* Impairs Inosine RNA Modifications in Mouse Oocytes

**DOI:** 10.3390/ijms22031191

**Published:** 2021-01-26

**Authors:** Pavla Brachova, Nehemiah S. Alvarez, Lane K. Christenson

**Affiliations:** Department of Molecular and Integrative Physiology, University of Kansas Medical Center, Kansas City, KS 661601, USA; nalvarez@kumc.edu

**Keywords:** inosine RNA modifications, oocyte, RNA decay, CCR4-NOT, GV-to-MII transition, polysome, translation, ADAR

## Abstract

Mammalian oocytes must degrade maternal transcripts through a process called translational mRNA decay, in which maternal mRNA undergoes translational activation, followed by deadenylation and mRNA decay. Once a transcript is translationally activated, it becomes deadenylated by the CCR4-NOT complex. Knockout of CCR4-NOT Transcription Complex Subunit 6 Like (*Cnot6l*), a deadenylase within the CCR4-NOT complex, results in mRNA decay defects during metaphase I (MI) entry. Knockout of B-cell translocation gene-4 (*Btg4*), an adaptor protein of the CCR4-NOT complex, results in mRNA decay defects following fertilization. Therefore, mechanisms controlling mRNA turnover have significant impacts on oocyte competence and early embryonic development. Post-transcriptional inosine RNA modifications can impact mRNA stability, possibly through a translation mechanism. Here, we assessed inosine RNA modifications in oocytes, eggs, and embryos from *Cnot6l^-/-^* and *Btg4^-/-^* mice, which display stabilization of mRNA and over-translation of the stabilized transcripts. If inosine modifications have a role in modulating RNA stability, we hypothesize that in these mutant backgrounds, we would observe changes or a disruption in inosine mRNA modifications. To test this, we used a computational approach to identify inosine RNA modifications in total and polysomal RNA-seq data during meiotic maturation (GV, MI, and MII stages). We observed pronounced depletion of inosine mRNA modifications in samples from *Cnot6l^-/-^*, but not in *Btg4^-/-^* mice. Additionally, analysis of ribosome-associated RNA revealed clearance of inosine modified mRNA. These observations suggest a novel mechanism of mRNA clearance during oocyte maturation, in which inosine-containing transcripts decay in an independent, but parallel mechanism to CCR4-NOT deadenylation.

## 1. Introduction

Growing mammalian oocytes transcribe and store transcripts that support meiotic maturation, fertilization, and early embryo development [1,2,3,4]. The shift of maternal to zygotic control of development is termed the maternal to zygotic transition (MZT), and occurs in the absence of transcription [4,5]. As oocytes resume meiosis, a large-scale, but selective wave of maternal transcripts are targeted for translation and subsequent degradation [6,7,8] in a process called translational mRNA decay. During meiotic resumption, dormant maternal mRNA with short poly(A) tails (20–40 nucleotides) undergo polyadenylation (80–250 nucleotides), in a mechanism controlled by the MAPK pathway, which activates translation of specific transcripts [9]. Translational activation is transient because the polyadenylated transcripts are quickly targeted for mRNA decay [10]. By the 2-cell stage of mouse early embryogenesis, approximately 90% of maternal mRNA are degraded [6]. The activation of translation requires specific 3′ untranslated region (UTR) elements, such as polyadenylation signals and cytoplasmic polyadenylation elements [11,12]. CCR4-NOT is the major, multi-subunit deadenylase complex involved in poly(A) shortening [13]. A functional CCR4-NOT complex is essential for the proper decay of maternal oocyte mRNA, and for the development of a fertilized oocyte into an embryo [14,15,16,17]. 

An understanding of the individual components of the CCR4-NOT complex has illuminated specific features of mRNA transcripts leading to their specific and selective decay. The ribonuclease CCR4-NOT Transcription Complex Subunit 6 Like (*Cnot6l*) is uniquely expressed in mouse oocytes, and regulates deadenylation of transcripts undergoing translational mRNA decay during oocyte maturation [14,15,18]. *Cnot6l^-/-^* mice are severely subfertile due to the over-translation of undegraded maternal transcripts, resulting in spindle defects during the completion of meiosis [14,15]. The CCR4-NOT deadenylase complex also relies on the adaptor RNA binding proteins, ZFP36 Ring Finger Protein Like 2) (ZFP36L2) and B-cell translocation gene-4 (BTG4), to associate with cytoplasmic transcripts [17,19,20,21,22,23]. *Zfp36l2^-/-^* mice are infertile due to decreased oocyte numbers as well as decreased rates of oocyte maturation [17,19]. *Btg4^-/-^* mice are also infertile due to embryonic arrest at the 2-cell and 4-cell stages [14,16]. Cumulatively, these models demonstrate that the translational mRNA decay machinery is necessary to establish the correct dosage of RNA during the MZT, and dysfunction or loss of critical components within the mRNA degradation pathway leads to infertility. 

Previous work has established that RNA modifications can impact mRNA stability and have a role in maternal mRNA clearance [24]. Inosines are a frequent RNA modification generated by the deamination of adenosine and catalyzed by a family of conserved double stranded RNA (dsRNA) binding proteins, adenosine deaminases acting on RNA (ADARs) [25,26,27]. *Adar*/ADAR is the predominant adenosine deaminase detected in mouse GV oocytes and MII eggs, and *Adarb2*/ADAR2 is also present [28]. Although inosine post-transcriptional RNA modifications can impact mRNA stability through a translation mechanism [29], the relationship of inosines and RNA stability in mouse oocytes has not yet been demonstrated. Since the *Cnot6l^-/-^* and *Btg4^-/-^* mice result in stabilization of mRNA and over-translation of stabilized transcripts, we hypothesized that in these mutant backgrounds, we would observe an increase in inosine modifications in mRNA. Surprisingly, we observed a significant decrease in inosine RNA modifications in oocytes, eggs, and zygotes from *Cnot6l^-/-^*, but not in *Btg4^-/-^* mice, suggesting that in the absence of *Cnot6l*, mRNA degradation of inosine-modified transcripts is still occurring, while unmodified transcripts are stabilized. Our data suggest novel components of translational mRNA decay during oocyte maturation.

## 2. Results

### 2.1. Inosine RNA Modifications Are Lost in Oocytes from Cnot6l^-/-^ Mice

Inosine RNA modifications are identified by comparing the transcriptome to the genome to identify A-to-G mismatches, as we and others have shown [28,30,31,32]. Using our computational approach [28], we identified transcriptome-wide inosine RNA modifications (Appendix A) in previously reported RNA-seq datasets [14] from wild-type (WT), *Cnot6l^-/-^*, and *Btg4^-/-^* oocytes and eggs at 0, 8, and 16 h after in vitro maturation. These three time points correspond to the GV, MI, and MII oocyte and egg stages, respectively. The number of unique transcripts with inosine RNA modifications in *Cnot6l^-/-^* samples (134 ± 28 in GV, 159 ± 0 in MI, 163 ± 3 in MII, and 160 ± 7 in zygotes (Zyg); Mean ± SEM) were significantly lower than both WT samples and *Btg4^-/-^* samples at all oocyte/egg/embryo stages (WT: 1750 ± 337 in GV, 999 ± 109 in MI, 1132 ± 213 in MII, and 1311 ± 562 in Zyg; *Btg4^-/-^*: 1749 ± 197 in GV, 1412 ± 30 in MI, 1579 ± 103 in MII, and 1441 ± 200 in Zyg; one-way ANOVA; Figure 1A). No difference in number of transcripts with inosine RNA modifications was detected between WT and *Btg4^-/-^* samples at any stage (Figure 1A). To confirm that the reduction in transcripts with inosine RNA modifications in *Cnot6l^-/-^* samples was not due to a differential pattern of RNA expression, we identified inosine modifications only in transcripts found in all samples across all groups (8686 transcripts, total RNA). The results (Figure 1C), indicate that the loss of inosine RNA modification was not due to differential gene expression. 

### 2.2. Inosine RNA Modifications Are Enriched in Ribosome-Associated mRNA

Our previous study linked inosine RNA modifications to the process of translation through RNA modifications of codons [28]. To test if the accumulated and over-translated maternal transcripts found in oocytes/eggs of *Cnot6l^-/-^* mice were enriched in inosine RNA modifications, we cataloged inosine RNA modifications in ribosome-associated mRNA (i.e., mRNA bound by multiple ribosomes) at the GV, MI, and MII stages in WT and *Cnot6l^-/-^* oocytes/eggs. The polysome RNA-seq datasets we analyzed were prepared by isolating mRNA bound by multiple ribosomes (i.e., the polysome) from groups of 500 whole oocytes (stages: GV, MI, MII). Polysome RNA-seq was not performed on Btg4^-/-^ oocytes or eggs [14]. In the normalized ribosome-associated mRNA of WT GV oocytes, the number of inosine-modified transcripts decreased during oocyte maturation, possibly indicating clearance of inosine-modified transcripts (Figure 1B). Conversely, in *Cnot6l^-/-^* oocytes/eggs, significantly fewer transcripts were inosine-modified at the GV stage, and the number of inosine-modified transcripts increased, rather than decreased during oocyte maturation (Figure 1B). Even among commonly expressed transcripts (6006 transcripts, ribosome-associated transcripts), the pattern was similar (Figure 1D). Our data show that globally, the total RNA fraction of *Cnot6l^-/-^* samples have fewer inosine modified transcripts, however, these modified transcripts are still detected in the ribosome-associated fraction. 

### 2.3. Pattern of Inosine RNA Modifications in Total and Ribosome-Associated mRNA

In addition to having the fewest inosine modified transcripts, samples from *Cnot6l^-/-^* mice had the lowest proportion of the transcriptome that was modified (GV: 1.0 ± 0.3%, MI: 1.2 ± 0.02%, MII:1.2 ± 0.03%, Zyg: 1.3 ± 0.05%), compared to WT samples (GV: 14.4 ± 2.6%, MI: 8.5 ± 0.7%, MII: 10.1 ± 1.7%, Zyg: 10.9 ± 4.3%) and *Btg4^-/-^* samples (GV: 13.4 ± 1.2%, MI: 10.6 ± 0.3%, MII: 13.0 ± 0.7%, Zyg: 12.0 ± 1.4%; Χ2 *p* < 0.5; Figure 2A). Among ribosome-associated mRNA, a lower proportion of transcripts were modified in *Cnot6l^-/-^* oocytes/eggs (GV: 6.1 ± 0.2%, MI: 9.7 ± 0.2%, MII: 8.7 ± 0.4%,), compared to WT oocytes/eggs (GV: 16.4 ± 0.3%, MI: 10.6 ± 0.2%, MII: 9.5 ± 0.1%, Figure 2B). Comparing the number of identified inosine RNA modifications per transcript, on average, WT and *Btg4^-/-^* samples had similar patterns, while *Cnot6l^-/-^* samples contained transcripts with fewer modifications per transcript (blue and green lines, Figure 2C). However, within ribosome-associated mRNA, the pattern of inosine RNA modifications per transcript was similar (Figure 2C). To further characterize the nature of inosine RNA modifications in samples from WT, *Cnot6l^-/-^*, and *Btg4^-/-^* mice, the location of modifications within protein-coding genes was determined by annotating the location of each inosine RNA modification (5′ UTR, coding, intron, and 3′ UTR). Overall, the distribution of inosine RNA modifications is similar between all stages of oocyte maturation in WT and *Btg4^-/-^* mice, with the majority of modifications occurring in the CDS and 3′ UTR. Conversely, samples from *Cnot6l^-/-^* mice display a higher proportion of intron inosine RNA modifications (Figure 2D). Within transcripts from ribosome-associated mRNA, the pattern of inosine RNA modifications was similar (Figure 2D).

### 2.4. Consequences of Coding Sequence Inosine RNA Modifications in Mouse Oocytes, Eggs, and Embryos

To understand the consequence of inosine RNA modifications, if any, on the protein coding capacity of mRNA transcripts, we used Ensembl Variant Effect Predictor (VEP) [33] to identify the synonymous and nonsynonymous substitutions (Figure 3A). Among the nonsynonymous substitutions, altered stop codons (stop loss, stop gain, or stop retained) made up less than 0.3% of coding sequence modifications. Therefore, stop codon substitutions were not included in further analyses. 

The majority of coding region inosine RNA modifications observed resulted in synonymous substitutions in WT and *Btg4^-/-^* samples [28]. Conversely, in samples from *Cnot6l^-/-^* mice, the pattern of inosine RNA modifications was reversed, with more than 60% exhibiting nonsynonymous substitutions (Figure 3A). Inosine RNA modified transcripts in the ribosome-associated mRNA had a similar pattern between WT and *Cnot6l^-/-^* samples (Figure 3B). In order to predict the consequence of the amino acid substitutions, a computational tool, Sorting Intolerant From Tolerant (SIFT) that predicts the effects of amino acid substitution on protein function was used [34]. SIFT analysis showed that inosine RNA modifications observed in WT and *Btg4^-/-^* samples exhibited greater levels of tolerated amino acid substitutions when compared to the samples from *Cnot6l^-/-^* mice (Figure 3C). Again, inosine RNA modified transcripts in the ribosome-associated mRNA had a similar pattern of tolerated amino acid substitutions between WT and *Cnot6l^-/-^* samples (Figure 3D). When considering the number of tolerated or deleterious substitutions, *Cnot6l^-/-^* samples have reduced numbers in total RNA (Figure 3E), but in the ribosome-associated transcripts, only GV *Cnot6l^-/-^* oocytes had significantly fewer tolerated substitutions (2-way ANOVA, *p* < 0.05, Figure 3F). In summary, the total RNA fraction in *Cnot6l^-/-^* samples had a higher proportion of inosine RNA modifications within the coding sequencing that resulted in potentially deleterious nonsynonymous substitutions. However, within the ribosome-associated RNA, similar levels of synonymous and nonsynonymous and tolerated and deleterious substitutions were observed between WT and *Cnot6l^-/-^*. These results suggest that translated mRNA in WT and *Cnot6l^-/-^* samples share a similar capacity for inosine induced protein recoding through alterations to codon usage. 

### 2.5. Inosine RNA Modifications Are Enriched at the Wobble Position in Ribosome-Associated RNA

The abundance of inosine modifications resulting in synonymous substitutions in WT and *Btg4^-/-^* samples led us to investigate the potential effects of inosine RNA modifications on codon usage. We first determined the number of inosine RNA modifications that occur in the 34 different codons that contain an adenosine, excluding stop codons. We found in WT and *Btg4^-/-^* samples, certain codons contained more inosine RNA modifications: AAA, ACA, CAA, CCA, GAA, and GCA (Figure 4A). In *Cnot6l^-/-^* samples, inosine-modified codons were significantly reduced compared with WT and *Btg4^-/-^* samples (Figure 4A). Additionally, there were no enriched modified codons detected in *Cnot6l^-/-^* samples (Figure 4A). Within transcripts from ribosome-associated mRNA of both WT and *Cnot6l^-/-^* samples, we detected similar codons enriched in inosine RNA modifications (Figure 4B). Further analysis indicated that inosine modifications were enriched at the wobble position in codons with multiple adenosines in total RNA of WT (Appendix A) and *Btg4^-/-^* samples (Appendix A), but not in *Cnot6l^-/-^* samples (Appendix A). Conversely, both WT and *Cnot6l^-/-^* ribosome-associated mRNA exhibited the wobble position enhancement (Appendix A). The wobble position RNA modification enrichment, across all codons, was not present in *Cnot6l^-/-^* samples in total RNA, in contrast to WT and *Btg4^-/-^* samples, while the ribosome-associated mRNA retained wobble position inosine RNA modifications across all genotypes (Figure 5A,B). 

### 2.6. Efficiency of Inosine RNA Modifications is Highest in Ribosome-Associated mRNA

Transcriptome-wide inosine RNA modification efficiency at various codons remained around 50% throughout oocyte maturation in WT samples (Figure 6A, Appendix A shows raw counts). Conversely, *Cnot6l^-/-^* samples displayed defects in inosine RNA modification efficiency in total RNA at all stages of oocyte maturation (Figure 6A, Appendix A). An examination of inosine RNA modification efficiency in *Btg4^-/-^* samples revealed a reduction of modification efficiency of select codons (Appendix A). The inosine RNA modification efficiency of zygotes decreased only in the *Cnot6l^-/-^* background (Appendix A). The inosine RNA modification efficiency reached almost 100% within transcripts of ribosome-associated mRNA of WT samples, and then decreased during oocyte maturation (Figure 6B and Appendix A). *Cnot6l^-/-^* samples also displayed almost 100% efficiency of particular codons (AAA, AAC, AAT, AGA, CAA, CAG, CCA, CGA, GTA, TTA), however, other codons were not modified at all (AAG, ACG, ACT, AGG, ATT, GAC, GAG, GAT), as seen in Figure 4. During oocyte maturation, the inosine RNA modification efficiency was completely lost at a majority of codons in *Cnot6l^-/-^* MI and MII oocytes (Figure 6B and Appendix A). Overall, during oocyte maturation, the inosine RNA modification efficiency is almost 100% in ribosome-associated mRNA, and decreases during maturation. At the MII stage of WT eggs, there is a positive correlation of RNA modification efficiency and RNA abundance (TPM), (Pearson correlation: 0.18, *p* < 0.05). Conversely, at the MII stage of *Cnot6l^-/-^* eggs, there is a negative association of RNA modification efficiency and RNA abundance (Pearson correlation: −0.13, *p* < 0.05). Thus, in the *Cnot6l^-/-^* oocytes/eggs where mRNA is stabilized, the inosine modified mRNA are degraded, while the unmodified mRNA are stabilized.

## 3. Discussion

Translational mRNA decay is the predominant mechanism of maternal mRNA clearance in mammalian oocytes. During oocyte maturation, mRNA are recruited for translation and subsequent degradation through interactions with components of the CCR4-NOT complex [14,18]. The maternal-to-zygotic decay machinery component CNOT6L plays an important role in maternal mRNA decay during MI [14,15,18], while the adaptor factor BTG4 plays an important role in degradation of maternal mRNA after ovulation and prior to fertilization [16,21]. As such, selective decay of transcripts allows for specific control of events occurring during the MZT, necessary for early embryo development. 

An important component of translational mRNA decay is the rate of translation, which is influenced by codon composition, a phenomenon known as codon optimality [35,36,37,38,39,40]. Codons that are nonoptimal slow the ribosome, and experimental evidence indicates that this increases mRNA degradation through CCR4-NOT [37]. It has recently been reported that inosine RNA modifications in the coding sequence of mRNA lead to ribosome stalling [29]. In a previous report [28], and now here in WT and *Btg4^-/-^* samples, we identified inosine RNA modifications specifically enriched at the codon wobble position in samples. Based on our finding and others, we hypothesized that inosine RNA modifications within the coding sequence could alter mRNA stability, potentially through translational mRNA decay. To test this hypothesis, we analyzed previously generated RNA-seq data for *Cnot6l^-/-^* and *Btg4^-/-^* samples, which are known to exhibit dysfunctional translational mRNA decay [14]. The nearly complete loss of inosine RNA modified transcripts in the *Cnot6l^-/-^* samples was unexpected, as we originally hypothesized increased modified transcripts would result from hyperstabilized mRNA as previously established [14]. The loss of inosine RNA modifications was not associated with a loss of Adar expression (Appendix A), but we have not evaluated ADAR protein levels in *Cnot6l^-/-^* samples, which is an important consideration. *Btg4^-/-^* samples, on the other hand, have inosine RNA modification profiles similar to that of WT samples. Both CNOT6L and BTG4 accumulate during oocyte maturation and trigger maternal mRNA decay, but their knockouts result in different oocyte phenotypes: *Cnot6l^-/-^* oocytes arrest at prometaphase I, while *Btg4^-/-^* oocytes are able to complete meiotic maturation and do not arrest until fertilization [14]. 

CNOT6L and BTG4 degradation pathways act on similar substrate pools during oocyte maturation, albeit with different timing [14]. We identified inosine modified transcripts common between all genotypes, and observed that *Cnot6l^-/-^* samples had significantly fewer modified transcripts compared to WT and *Btg4^-/-^* samples (Figure 1C). We consider three possible mechanisms that can explain our results. First, CNOT6L and ADAR could interact in oocytes/eggs, facilitating inosine RNA modifications. Biological evidence for such a complex is lacking. Recent mass spectrometry approaches to identify ADAR associated proteins that regulate inosine RNA modifications have not identified CCR4-NOT complex components [41]. Rather, several components of the ribosome were identified directly associated with ADAR [41].

The second explanation for reduced inosine RNA modifications in *Cnot6l^-/-^* samples is that Adar expression is reduced, thus reducing inosine RNA modifications. We have previously shown that the inosine modification signature in oocytes/eggs (as determined by A-to-G substitutions in RNA-seq data) is linked to the catalytic activity of ADAR [28]. Within the ribosome-associated mRNA, the number of inosine modifications per transcript, codon wobble position modifications, and efficiency of modifications revealed that both WT and *Cnot6l^-/-^* samples had similar levels of inosine RNA modifications at the GV stage (Figure 3C, Figure 5A,B and Figure 6B). Additionally, both WT and *Cnot6l^-/-^* samples had similar levels *Adar* mRNA expression (Appendix A). Based on the well-established catalytic signature of ADAR present in RNA-seq from GV oocytes of ribosome-associated mRNA in both WT and *Cnot6l^-/-^* samples, we do not consider a reduction of ADAR abundance to be a likely explanation for the decrease in inosine RNA modifications present in *Cnot6l^-/-^* samples; however, further experimentation would be needed to test ADAR levels in the absence of CNOT6L.

A third possible mechanism to explain the reduction in inosine RNA modifications in *Cnot6l^-/-^* samples is that CNOT6L and inosine RNA modifications both contribute to translational mRNA degradation, through two independent, but parallel pathways. Experimental evidence indicates that codon composition can lead to slowing of the ribosome, which increases mRNA degradation through CCR4-NOT [42,43]. It has been reported that inosine RNA modifications within codons of mRNA can lead to ribosome stalling [29]. Licht et al. used an in vitro system to show that single inosine modifications in codons were sufficient to induce ribosome stalling and truncation of peptides, albeit at a low frequency of 5%. Multiple inosines within codons caused ribosome stalling and peptide truncation at a frequency of 30%. Additionally, Licht et al. [29] used ribosome profiling data to show that, in vivo, ribosomes accumulate at single inosine containing codons. Their results indicated that within cells, a single inosine residue within a codon was sufficient to stall or slow ribosomes. 

Here, we show that ribosome-associated mRNA have similar inosine enrichment to the wobble position of codons in both WT and *Cnot6l^-/-^* samples, confirming observations in our previous report (Figure 6B) [28]. Furthermore, in the ribosome-associated fraction of GV oocytes from WT and *Cnot6l^-/-^* mice, inosine RNA modification efficiency was above 60%, and in some cases almost 100% for the codons analyzed (Figure 6B). In comparison, in the total RNA fraction, inosine modification efficiency was approximately 50% (Figure 6A). In the ribosome-associated fraction of WT samples, progression into MI phase was marked by a reduction in inosine RNA modification efficiency (Figure 6B), but in *Cnot6l^-/-^* samples, there is an almost complete loss of inosine RNA modification efficiency of codons (Figure 6B). A major wave of translational recruitment and mRNA decay occurs during the GV–MI transition [11,14]. In the absences of CNOT6L, mRNA is stabilized from MI through MII stages, presumably due to defects in translational mRNA decay [14]. The stabilization of MI mRNA coincides with a decrease in inosine modifications at codons from the polysome fraction (Figure 6B). The decrease in translation mRNA decay in MI oocytes and MII eggs, coupled with the loss of mRNA with inosine modified codons, suggests a secondary mechanism for mRNA clearance. An important consideration for our analysis is the method employed to isolate polysomes from the RNP fraction [11]. The data utilized in our analysis were derived from the polysomal fraction, excluding the RNP fraction [14]. In oocytes, polysome composition is made up of transcripts that are shuttled from the RNP fraction [11]. It is conceivable that inosine RNA modifications could occur within the RNP fraction prior to mRNA being transferred to the polysome fraction. Further experimentation is needed to determine if the RNP fraction from *Cnot6l^-/-^* oocytes shows similar reduction in inosine modifications as the polysome fraction. 

It is possible that CNOT7 or CNOT8 could be compensating for the absence of CNOT6L to mediate the decay of inosine containing transcripts. We observed that among common ribosome-associated inosine modified transcripts, in WT oocytes/eggs, the abundance of mRNA has a positive correlation with inosine modification efficiency (0.18, *p* < 0.05, Pearson), while in *Cnot6l^-/-^* oocytes/eggs, we observed a negative correlation (−0.13, *p* < 0.05, Pearson). If CNOT7 or CNOT8 were compensating for an absence of CNOT6L, we would predict a similar correlation of mRNA abundance to inosine modification efficiency in both WT and *Cnot6l^-/-^* samples because the inosine modified transcripts would still be degraded in the absence of CNOT6L. However, we observed the opposite; mRNA stabilized in the absence of CNOT6L is targeted for degradation if it contains inosine modified codons. 

Another important consideration of our results is that the codon wobble position enrichment of inosine is absent in the total RNA fraction *Cnot6l^-/-^* oocytes/eggs (Figure 5A,B). Furthermore, the number of transcripts with inosine RNA modifications is significantly reduced in the total RNA fraction of *Cnot6l^-/-^* samples when compared to WT (Figure 1A). We reason that one explanation for the difference between the amount of RNA transcripts with inosine RNA modifications between total RNA and ribosome-associated mRNA fractions is due to the number of samples used and the subsequent enrichment of ribosome fraction (total RNA: 10 oocytes/eggs/zygotes; ribosome-associated RNA: 500 oocytes/eggs). The inosine modified transcripts in the ribosome fraction are also present in the total RNA, albeit at a smaller proportion, and enrichment allows for their detection. Additionally, the observed reduction of inosine RNA modifications in *Cnot6l^-/-^* samples within the total RNA fraction may be due to two factors: overall increased mRNA stability and selective decay of inosine modified mRNA. In *Cnot6l^-/-^* samples, mRNA is globally stabilized, and we hypothesize that inosine modified RNA is targeted for decay. We reason that the stabilization of mRNA increases the mRNA complexity of the total RNA fraction, thus reducing the sensitivity of RNA-seq to detect inosine modified RNAs. The same phenomenon is present in the ribosome-associated fraction of *Cnot6l^-/-^* samples, but it is offset by the ribosome enrichment procedure. Based on our results, we hypothesize that CNOT6L dependent mRNA decay and inosine RNA modifications share a common component: translation. Our results suggest a novel mechanism of mRNA clearance during oocyte maturation, in which inosine-containing transcript decay is uncoupled from CCR4-NOT deadenylation. Further experiments will reveal the impact of inosine RNA modifications on maternal mRNA decay.

## 4. Materials and Methods 

### 4.1. Sources of GV and MII RNA-Seq Datasets

WT GV, MI, MII, and Zygote RNA-seq data, and GV, MI, MII, and Zygote RNA-seq data from *Cnot6l^-/-^* and *Btg4^-/-^* knockout mice dataset were downloaded from the short read archive (SRA) PRJNA486094 [14]. The authors of this study generated library preparations from total RNA isolated from groups of 10 oocytes, eggs, or embryos, all on the C57B6 background. The polysome RNA-seq datasets we analyzed were prepared by isolating mRNA bound by multiple ribosomes (i.e., the polysome) from groups of 500 whole oocytes (stages: GV, MI, MII). Two replicates were generated for library preparation. Each library preparation had mCherry spike-in added before SMART-Seq2 cDNA synthesis for normalization. Barcoded libraries were pooled and sequenced on an Illumina HiSeq X Ten platform with 150 bp paired-end reads [14]. 

### 4.2. Identification and Consequence Analysis of Inosine RNA Modifications

We identified putative inosine RNA modifications utilizing a combination of the following software: HISAT2 aligner, the Genome Analysis Tool Kit (GATK), and Ensembl Variant Effect Predictor (VEP) [28,33,44,45]. Inosines appear as A-to-G substitutions when comparing RNA-seq data to a reference genome [46]. HISAT2 was used to map raw RNA-seq reads to a *Mus musculus* index built from GRCm38 containing common SNP annotation from dbSNP. 

Default settings for HISAT2 were used for each type of library aligned (i.e., single-end, paired-end, or stranded) [45]. Prior to alignment, fastq files were checked for adaptors and trimmed if necessary, using Trimmomatic [47]. The default alignment settings for HISAT2 will report at most 10 valid primary alignments; we did not increase the amount of multimapping allowed. Mismatches in the alignment were subtracted from the alignment score, reads that fell below the minimum alignment score were not reported. RNA/DNA differences were called using the Genome Analysis Toolkit (GATK) RNASeq variant pipeline with modifications [44]. The program elprep was used to sort, mark duplicates, and index RNA-seq reads. Known SNPs were filtered out using the Mouse Genomes Project database, Wellcome Trust Sanger Institute mouse strains [48]. RNA–DNA mismatches identified by HISAT2, regardless if stranded mode was used, and GATK were reported in the sense orientation; therefore, strand information of the variant is inferred from the gene model using Ensembl VEP [33]. VCF files were filtered on allelic depth (AD > 0) and used as input for VEP [33]. VEP was used to identify inosine modified transcripts, categorize the location within the transcript, and determine the consequence of inosines on coding capacity. Only inosine sites occurring in transcripts with TPM (transcripts per million) ≥ 1 and having an AD > 0 were reported. TPM values for total and polysome RNA were calculated using Kallisto (v0.45.1) and GCRm38 transcriptome reference and normalized to mCherry spike in [49]. A transcript was considered to be inosine modified if it contained at least one inosine site. R statistical computation software with the following packages was used to parse VEP output: sleuth, biomaRt, dplyr, plyr, AnnotationFuncs, org.Mm.eg.db, ggplot2 [50,51,52,53].

## Figures and Tables

**Figure 1 ijms-22-01191-f001:**
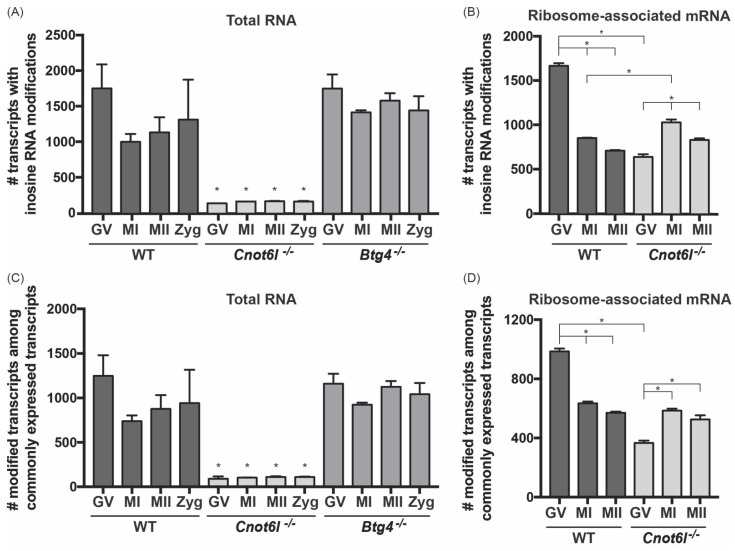
Inosine RNA modifications occur in transcripts enriched in the ribosome-associated RNA. (**A**,**B**) The number of unique inosine modified transcripts identified in wild-type (WT), CCR4-NOT Transcription Complex Subunit 6 Like (*Cnot6l*)^-/-^, and B-cell translocation gene-4 (*Btg4*)^-/-^ GV, MI, MII, and Zyg in total RNA (**A**) and ribosome-associated mRNA (**B**). (**C**,**D**) Among common transcripts present in all samples, the number of unique inosine modified transcripts identified in WT, *Cnot6l*^-/-^, and *Btg4*^-/-^ GV, MI, MII, and Zyg in total RNA (**C**) and ribosome-associated mRNA (**D**). *Means ± SEM within panels A and B are different (*p* < 0.05).

**Figure 2 ijms-22-01191-f002:**
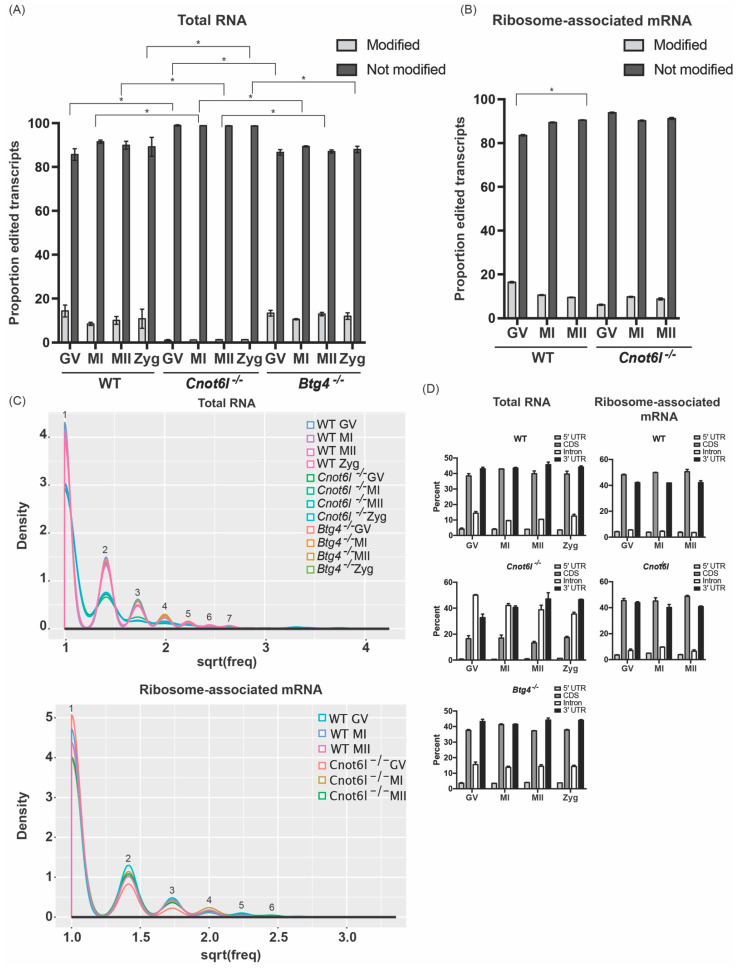
Distinct pattern of inosine RNA modifications from WT, *Cnot6l*^-/-^, and *Btg4*^-/-^ oocytes, eggs, and zygotes. (**A**,**B**) Proportion of the transcriptome (percentage) that contains inosine RNA modifications in total RNA (**A**) and ribosome-associated mRNA (**B**). (**C**) Number of inosine RNA modified transcripts exhibiting one or multiple inosines per transcript in total RNA and ribosome-associated RNA. Numbers above line indicate the number of inosines/transcript. (**D**) Number of inosine RNA modifications within specific regions (5′ UTR, CDS, intron, and 3′ UTR) in mRNA in total RNA and ribosome-associated RNA. *Means ± SEM within panels A and B are different (*p* < 0.05); significance was determined using X2 tests. Only transcripts with TPM ≥ 1 were analyzed.

**Figure 3 ijms-22-01191-f003:**
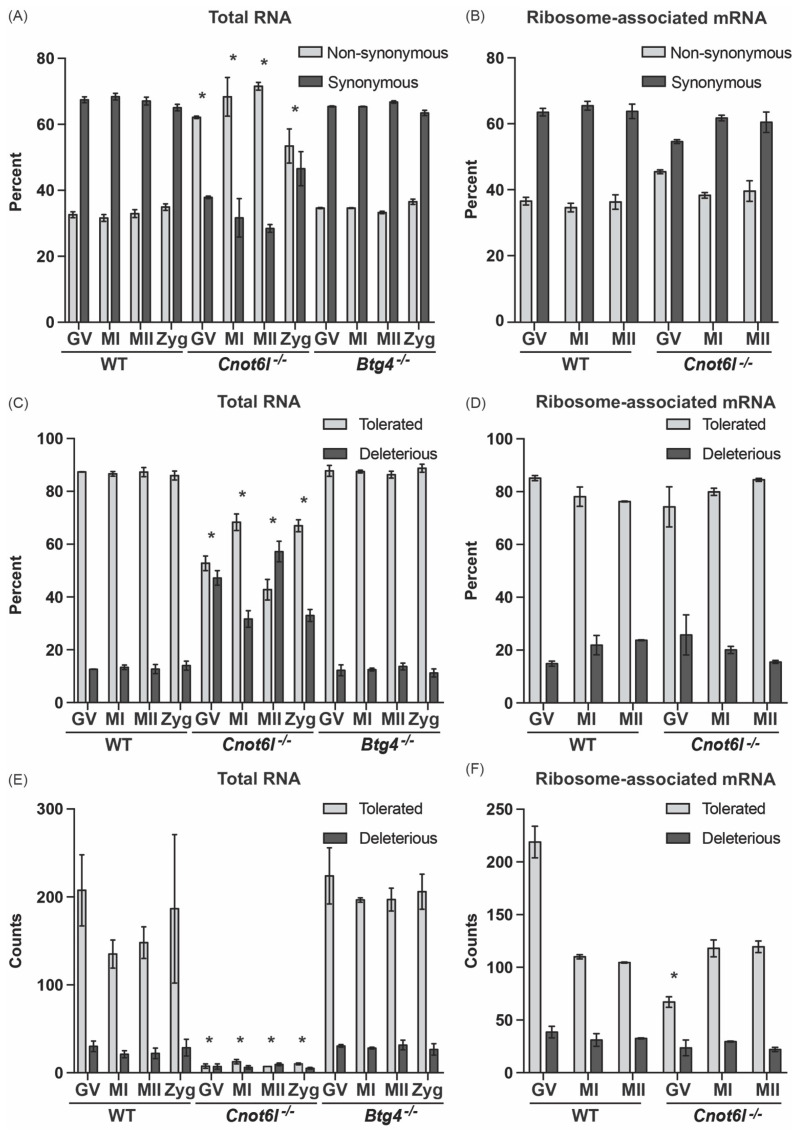
Consequences of inosine RNA modifications within CDS from WT, *Cnot6l*^-/-^, and *Btg4*^-/-^ oocytes, egg, and zygote transcripts. (**A**,**B**) Proportion of inosine modifications (percentage) in WT, *Cnot6l*^-/-^, and *Btg4*^-/-^ samples that result in nonsynonymous or synonymous changes was determined for all CDS inosine modified mRNA transcripts in total RNA (**A**) and ribosome-associated mRNA (**B**). (**C**,**D**) Proportion of tolerated and deleterious transcripts following Sorting Intolerant From Tolerant (SIFT) analysis of the inosine modified mRNA transcripts from WT, *Cnot6l*^-/-^, and *Btg4*^-/-^ samples in total RNA (**C**) or ribosome-associated mRNA (**D**). (**E**,**F**) Counts of tolerated and deleterious transcripts resulting from inosine RNA modifications from WT, *Cnot6l*^-/-^, and *Btg4*^-/-^ samples in total RNA (**E**) or ribosome-associated mRNA (**F**). *Means ± SEM within a panel are different (*p* < 0.05); significance was determined using Χ2 tests or 2-way ANOVA. Only transcripts with TPM ≥ 1 were analyzed.

**Figure 4 ijms-22-01191-f004:**
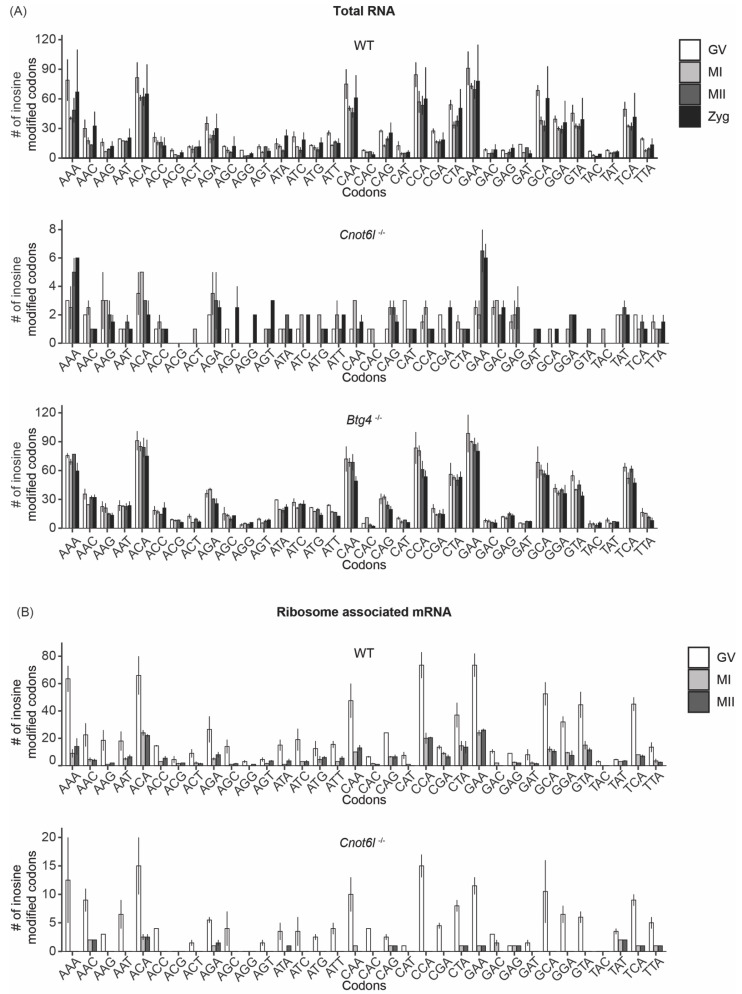
Inosines RNA modifications occur at specific codons. (**A**,**B**) Number of inosine RNA modifications in adenosine-containing codons in WT, *Cnot6l*^-/-^, and *Btg4*^-/-^ oocytes, eggs, and zygotes in total RNA. Only codons from transcripts with TPM ≥ 1 were analyzed.

**Figure 5 ijms-22-01191-f005:**
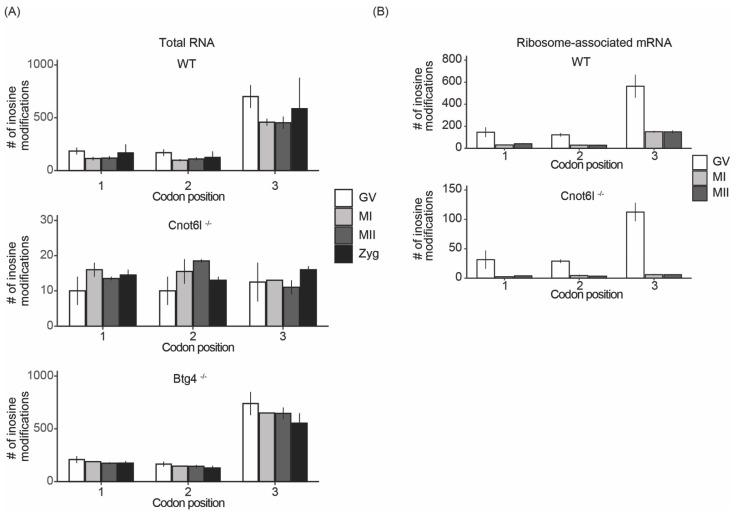
Inosines are enriched at the wobble position of codons in total RNA and ribosome-associated RNA. (**A**,**B**) Global frequency of inosine RNA modifications at the first, second, or third codon positions in WT, *Cnot6l*^-/-^, and *Btg4*^-/-^ samples in total RNA (**A**) and ribosome-associated RNA (**B**). Only codons from transcripts with TPM ≥ 1 were analyzed.

**Figure 6 ijms-22-01191-f006:**
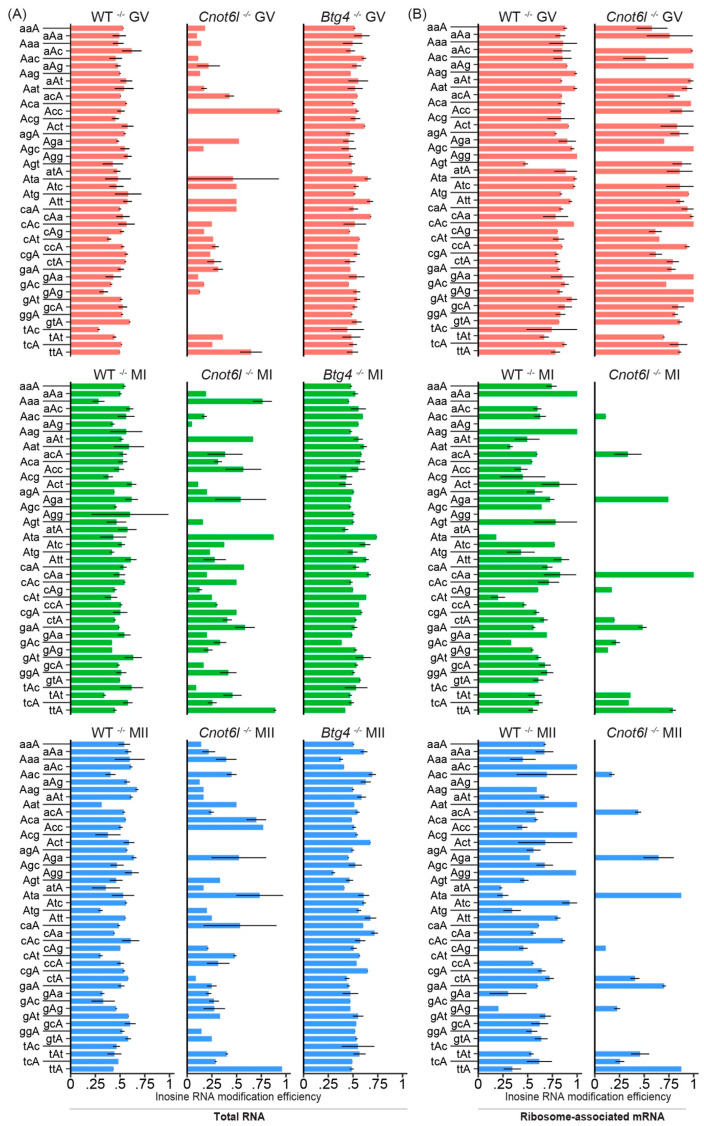
Inosine RNA modification efficiency of total and ribosome-associated mRNA. (**A**) Inosine RNA modification efficiency in total RNA from WT, *Cnot6l*^-/-^, and *Btg4*^-/-^ oocytes, eggs. (**B**) Inosine RNA modification efficiency in ribosome-associated mRNA from WT and Cnot6l^-/-^ oocytes, eggs. Capital letter denotes the position of the codon with an inosine RNA modification in both panels. Oocyte and egg stages are represented by red bars (GV), green bars (MI), and blue bars (MII).

## Data Availability

Not applicable.

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
