# Peer review of "Loss of Cnot6l Impairs Inosine RNA Modifications in Mouse Oocytes"

_ijms, 2021, doi:10.3390/ijms22031191_

Round 1

Reviewer 1 Report

Authors bioinformatically analyze inosine mRNA modification from available RNA-seq datasets generated from total mRNA or ribosomal fractions obtained from oocytes and zygotes with absence CNOT6. The study is descriptive and bringing novel insights to RNA metabolism in the cell.

Overall, despite of missing experimental validation of generated data I find that this study is of high interest for the field of molecular biology of the cell.

Please see comments to the manuscript below.

Introduction:

Authors should introduce ,,translational mRNA decay,, and ,, inosine modification,,.

L40 the reference (9) belongs to experimental downregulation of specific transcript/s.

L41-43 Idea presented is not applicable generally in the oocyte or cell.

Ref24 what do we know about ribosome staling and inosine mRNA modification and decay in oocyte?

Results:

Clearly define ribosome associated; polysomal pool (L24?), monosomal pool or both.

Clearly mention why Btg4-/- analyses are missing in the figures.

Fig 1B L97 Mention about decreased ribosome occupancy positively correlate with previous findings about decreased global translation. Does authors normalize data to total RNA amount?

Fig2 E) is missing

Fig4 Font should be modified for better reading.

Fig5 Increase font size in the graphs. Descriptions missing. Does authors apply some statistical analyses to present significances observed between genotypes of MI and MII stages (Fig5B)?

Fig S4 Hard to read the data in the graph. Present data different way (e.g. split, change depiction). How many Adar variants are expressed in the oocyte?

Discussion:

Heavy part of polysomal gradient might contain non-translating structures, e.g. mRNA-RNP, stalled ribosomes, ribosomal lattices. Could authors discuss possibility of mRNA modification in relation to the non-translating cytoplasmic structures.

Will be beneficial to discuss finding about loss of inosine RNA modification in the MI of Cnot7-/- genotype.

Graphical abstract would be beneficial for the paper.

L297 word repetition

Author Response

We appreciate the constructive critiques provided by the reviewers. We have addressed the concerns by expanding explanations and making corrections in the text. Please see our responses to specific questions below. 

Reviewer 1:

Authors bioinformatically analyze inosine mRNA modification from available RNA-seq datasets generated from total mRNA or ribosomal fractions obtained from oocytes and zygotes with absence CNOT6. The study is descriptive and bringing novel insights to RNA metabolism in the cell.

Overall, despite of missing experimental validation of generated data I find that this study is of high interest for the field of molecular biology of the cell.

Please see comments to the manuscript below.

Introduction:

  1. Authors should introduce ,,translational mRNA decay,, and ,, inosine modification,,.

    In lines 39-42 we have expanded on the explanation introducing translational mRNA decay, and in lines 65-70 we have expanded on the introduction of inosine RNA modifications.

  2. L40 the reference (9) belongs to experimental downregulation of specific transcript/s.

    We thank the review for catching this error and have replaced the citation with the correct one (line 43, Su et al. 2007).

  3. L41-43 Idea presented is not applicable generally in the oocyte or cell.

    We agree with the reviewer the following sentence was a general statement only applicable to a few transcripts. We have removed the sentence “Once transcripts undergo translational activation following the lengthening of their poly(A) tails, they are subject to decay by deadenylation [12].”

  4. Ref24 what do we know about ribosome staling and inosine mRNA modification and decay in oocyte?

    Reference 24 (Higuchi et al. 2000; Point mutation in an AMPA receptor gene rescues lethality in mice deficient in the RNA-editing enzyme ADAR2.) This study shows that ADAR2-/- mice have reduced inosine RNA modifications, are prone to seizures, and die young (median survival 5 days). The early lethality phenotype of loss of ADAR2 can be rescued by transgenic expression of edited GluR-BR alleles. This study did not do an in-depth analysis of translation dynamics or ribosome stalling, or even inosine RNA modifications. They only assessed known, common ADAR2 target sites. They also did not examine the oocyte or fertility. Our previous study (Brachova et al, 2019) assessed inosine RNA modifications in oocytes, and this current manuscript validates those findings. The relationship of ribosome stalling and inosine RNA modifications in oocytes has not been assessed. There are currently no datasets performing ribosome footprinting in oocytes, a valuable future direction we hope to pursue, as it will test the relationship of ribosome stalling and inosine RNA modifications. We have cited our study in the current manuscript in several locations documenting our findings in mouse oocytes (Lines 66, 78-79, 102, 152, 247, 274, 296). 

Results:

  1. Clearly define ribosome associated; polysomal pool (L24?), monosomal pool or both.

    The polysome RNA-seq dataset we analyzed was prepared by isolating mRNA bound by multiple ribosomes (i.e., the polysome) from groups of 500 whole oocytes (stages: GV, MI, MII) (Sha, et al. EMBO, 2018). We added these details to the methods (Lines 348-353) and in the results section (Lines 104-105).

  2. Clearly mention why Btg4-/- analyses are missing in the figures.

    Unfortunately the publication from which we obtained our data (Sha, et al. EMBO, 2018) did not perform polysome mRNA-sequencing on Btg4-/- oocytes. We have made this more clear in the results section (Lines 107-108).

  3. Fig 1B L97 Mention about decreased ribosome occupancy positively correlate with previous findings about decreased global translation. Does authors normalize data to total RNA amount?

    We have provided additional information for RNA normalization in the methods (378 ). As well as indicating the data from the polysome was normalized.

  4. Fig2 E) is missing.

    Fig2 does not have an E panel and no references to Fig. 2E were found in manuscript text.

  5. Fig4 Font should be modified for better reading.

    We have adjusted our figures to enhance readability. As a result we have split Fig4 into two separate figures (Fig4 and Fig5) and moved data into three supplemental figures (S2, S3, S4, S5).

  6. Fig5 Increase font size in the graphs. Descriptions missing. Does authors apply some statistical analyses to present significances observed between genotypes of MI and MII stages (Fig5B)?

    We have adjusted our figures to enhance readability. Fig5 is now Fig6. We moved zygote data to supplemental figure S6. Fig5 now allows a more direct inspection between genotypes and total and polysome fractions. We utilized a 2 way ANOVA to determine if there was a significant difference between genotypes and oocyte states. The statistical test results are provided in a supplemental data file.

  7. Fig S4 Hard to read the data in the graph. Present data different way (e.g. split, change depiction).

    The graph has been split as suggested to make visualization easier. We have also moved this figure to supplemental figure 5.

  8. How many Adar variants are expressed in the oocyte?

    We have previously published (Brachova, et al. BOR, 2019) that mouse GV oocytes and MII eggs contain high levels of ADAR1 p110 and moderate levels of ADAR2. (Line 64-66). 

Discussion:

  1. Heavy part of polysomal gradient might contain non-translating structures, e.g. mRNA-RNP, stalled ribosomes, ribosomal lattices. Could authors discuss possibility of mRNA modification in relation to the non-translating cytoplasmic structures.

    We have added to the discussion to address this interesting observation (lines 307-314).

    An important consideration is that for our analysis is the method employed to isolate polysomes from the RNP fraction (Chen et al. 2011). The data utilized in our analysis were derived from the polysomal fraction, excluding the RNP fraction (Sha et al. 2018). In oocytes, polysome composition is made up of transcripts that are shuttled from the RNP fraction (Chen et al. 2011). It is conceivable that inosine RNA modifications could occur within the RNP fraction prior to mRNA being transferred to the polysome fraction. Further experimentation is needed to determine  if the RNP fraction from Cnot6l-/- oocytes shows similar reduction in inosine modifications as the polysome fraction.

  2. Will be beneficial to discuss finding about loss of inosine RNA modification in the MI of Cnot7-/- genotype.

    We have added additional content to the discussion (lines 302-307).

    A major wave of translational recruitment and mRNA decay occurs during the GV–MI transition (Sha et al. 2018; Chen et al. 2011). In the absences of CNOT6L, mRNA is stabilized from MI through MII stages, presumably due to defects in translational mRNA decay (Sha et al. 2018). The stabilization of MI mRNA coincides with a decrease in inosine modifications at codons from the polysome fraction (Fig. 6b). The decrease in translation mRNA decay in MI and MII oocytes, coupled with the loss of mRNA with inosine modified codons suggests a secondary mechanism for mRNA clearance.

  3. Graphical abstract would be beneficial for the paper.

    Thank you for this suggestion. We have updated the manuscript to include a graphical abstract.

  4. L297 word repetition.

    Corrected

Reviewer 2 Report

The manuscript by Brachova et al. have demonstrated that inosine modifications in total and ribosome-associated RNAs are dramatically down-regulated in Cnot6l-/- oocytes and that it may account for a parallel mechanism to CCR4-NOT mediated deadenylation and decay of maternal transcripts in oocytes and early embryos in mice. 

The experiments are appropriately designed and results obtained were clearly demonstrated, however, there is a missing link in between the occurrence of RNA inosine modifications and its physiological impact in mouse oocyte biology.  This reviewer believes that this manuscript will be suitable for acceptance in the journal IJMS when its substantial revision been made successfully, how inosine modifications impact normal process of oocyte maturation, fertilization and early embryonic development will be of particular subject of inquiry.

Specific comments on Figure 1: Why numbers of transcripts with inosine modifications in ribosome-associated mRNAs of Cnot6l-/- (panel B) are larger than those in total RNAs of Cnot6l-/- (panel A)?

Author Response

The manuscript by Brachova et al. have demonstrated that inosine modifications in total and ribosome-associated RNAs are dramatically down-regulated in Cnot6l-/- oocytes and that it may account for a parallel mechanism to CCR4-NOT mediated deadenylation and decay of maternal transcripts in oocytes and early embryos in mice. 

  1. The experiments are appropriately designed and results obtained were clearly demonstrated, however, there is a missing link in between the occurrence of RNA inosine modifications and its physiological impact in mouse oocyte biology.  This reviewer believes that this manuscript will be suitable for acceptance in the journal IJMS when its substantial revision been made successfully, how inosine modifications impact normal process of oocyte maturation, fertilization and early embryonic development will be of particular subject of inquiry.

We agree that the physiological link between inosine RNA modifications and oocyte biology is missing, and is an important area of research within our laboratory. We are extremely interested in the impact of inosine mRNA modifications on oocyte maturation, fertilization, and embryo development. We thank the reviewer for noting the importance of these questions, however, this manuscript is specifically testing the effect of RNA stabilization (Cnot6l KO and Btg4 KO) on inosine RNA modifications, and the addition of animal models would be beyond the scope of our project. Furthermore, given the COVID-19 environment, our mouse facility was shut down and we have only recently begun to re-establish our mouse colonies.  

  1. Specific comments on Figure 1: Why numbers of transcripts with inosine modifications in ribosome-associated mRNAs of Cnot6l-/- (panel B) are larger than those in total RNAs of Cnot6l-/- (panel A)? 

In our discussion, we reason that one explanation for the difference between the amount of RNA transcripts with inosine RNA modifications between total RNA and ribosome-associated mRNA fractions is due to the number of samples used and the subsequent enrichment of ribosome fraction (total RNA: 10 oocytes/eggs/zygotes; ribosome-associated RNA: 500 oocytes/eggs). The inosine modified transcripts in the ribosome fraction are also present in the total RNA, albeit at a smaller proportion, and enrichment allows for their detection. Additionally, the observed reduction of inosine RNA modifications in Cnot6l-/- samples within the total RNA fraction may be due to two factors: overall increased mRNA stability and selective decay of inosine modified mRNA. In Cnot6l-/- samples, mRNA is globally stabilized, and we hypothesize that inosine modified RNA is targeted for decay. We reason that the stabilization of mRNA increases the mRNA complexity of the total RNA fraction, thus reducing the sensitivity of RNA-seq to detect inosine modified RNAs. The same phenomenon is present in the ribosome-associated fraction of Cnot6l-/- samples, but it is offset by the ribosome enrichment procedure. (Lines 325-344).

Round 2

Reviewer 2 Report

I understand that the addition of experimental data using animal models or other functional means would be beyond the author's scope in this study.